# Policy Driven Generative Adversarial Networks for Accented Speech Generation

## Abstract

In this paper, we propose the generation of accented speech using generative adversarial networks (GANs). Through this work we make two main contributions a) The ability to condition latent representations while generating realistic speech samples b) The ability to efficiently generate long speech samples by using a novel latent variable transformation module that is trained using policy gradients. Previous methods are limited in being able to generate only relatively short samples or are not very efficient at generating long samples. The generated speech samples are validated through a number of various evaluation measures viz, a Wasserstein-GAN critic loss and through subjective scores on user evaluations against a competitive speech synthesis baseline. The evaluations demonstrate that the model generates realistic long speech samples conditioned on accent efficiently.

## 1 Introduction

This work addresses the problem of synthesising speech data as a raw audio waveform using generative adversarial networks (GANs). While generative adversarial frameworks have been used in a variety of settings mainly for images, generating speech using GANs has been more challenging. Speech requires generating a continuous sequential waveform requiring explicit control on content generated. Adversarial training of a recurrent sequence is more challenging especially in the continuous setting. One way to solve this is by generating an output sample of a predefined fixed size. However, generating longer samples would not be feasible in this setting.

In this paper, we solve the problem by treating it as generation of a sequence of speech synthesis segments. A naive implementation of this would not allow for generation of a continuous speech output as they would be discontinuous segments. We address this problem by using recurrent units that constrain and condition latent variables used for generation. End-to-end gradient based training of this setup turns out to perform very badly on its own. We address this issue by making use of a policy gradient framework. This allows for generation of a long sequence of speech from a sequence of generators that have latent variables trained using policy gradients. A discriminator evaluates the full sequence to adversarially provide feedback regarding the realism of the generated audio samples.

We focus on the generation of accented speech samples. Generating accented speech is an important problem that could lead to 1) increased comprehension when the accent matches that of the listener (Ikeno & Hansen, 2007) and 2) better speech recognition for accented speech by augmenting data used to train speech recognition systems. The latent variables in our proposed model are conditioned for both accent and content. The accent part uses a discrete random variable conditioned using accent labels, which differs from the continuous content based random variable.

Our main contributions can be summarized as follows:

- We provide an efficient GAN based framework that generates speech using a segmented speech synthesis approach.
- An efficient policy gradient based approach for training that allows for training rich set of latent variables.
- Explicit conditioning on accent to enable generation of a variety of natural audio with the ability to control the accent.

- Generation of long speech signals using a sequence of generators that are jointly discriminated using a single discriminator.
- Thorough evaluation of the framework using a Wasserstein critic, accent discriminator and human evaluation studies.

## 2 RELATED WORK

Statistical parametric speech synthesis systems have been extensively studied in prior work and achieved initial success using hidden Markov model-based systems (Zen et al., 2009). Among neural network-based approaches, both deep feed-forward neural networks (Zen et al., 2013) and recurrent neural networks (Fan et al., 2014) have been successfully used in building statistical parametric speech synthesis systems. More recent approaches have explored the use of neural networks for the entire speech synthesis pipeline (Oord et al., 2016; Mehri et al., 2017; Wang et al., 2017; Arik et al., 2017). Our work is similar in principle to two neural speech generation systems, Wavenet (Oord et al., 2016) and SampleRNN (Mehri et al., 2017), that both present powerful generative models of audio and operate directly on raw audio waveforms. Both Wavenet and SampleRNN, however, are slower than our approach on account of their autoregressive properties. Our speed up in generating samples comes from not having to re-encode samples during generation and instead sampling from rich latent variables with the help of recurrent units trained using policy gradients.

This work focuses on generating accented speech samples. This could help improve listeners' comprehension of the underlying content (Ikeno & Hansen, 2007). Also, prior work has shown that listeners show a preference for synthesized speech clips generated in their own accents (Tamagawa et al., 2011; Schreitter & Krenn, 2014). Generating speech samples in varying accents could also potentially impact automatic speech recognition (ASR) systems. Variability in speech accents pose a significant challenge to speech recognition systems (Benzeghiba et al., 2007). The presence of accented speech samples during ASR training could help with building more accent-robust speech recognition systems. Prior work on accented speech synthesis has predominantly made use of HMM-based synthesis models (Tomokiyo et al., 2005; Karhila & Wester, 2011; García Lecumberri et al., 2014; Toman & Pucher, 2015). We provide the first end-to-end neural framework to generate accented speech.

Our proposed approach is an end-to-end paradigm that is based on GANs. GANs have been used for speech synthesis in combination with traditional statistical speech synthesis models (Kaneko et al., 2017; Yang et al., 2017) and for voice conversion where a WGAN objective was introduced to improve the quality of the generated samples (Hsu et al., 2017). SEGAN (Pascual et al., 2017) and FSEGAN (Donahue et al., 2017) are the only prior works we know of that use GANs in an end-to-end manner for speech generation. They propose speech enhancement systems that take raw audio waveforms as input and produce enhanced speech as output. SEGAN (Pascual et al., 2017) is one of the primary baseline systems we compare against.

Our work largely draws inspiration from the SeqGAN framework(Yu et al., 2017), which first introduced the idea of modeling the generator as an agent with a stochastic policy which is trained via policy gradients. In SeqGAN (Yu et al., 2017), discrete sequence generation is modeled within a GAN framework. The authors avoid obtaining generator gradients through a policy update step. As we obtain segments of speech synthesis, we also benefit by using a policy update. Further, work by Dai et al. (2017) have applied policy update through an actor-critic framework where the policy update for the intermediate steps are obtained by completing the action through a deterministic LSTM update. This model could be used with the updates being obtained through deterministic policy gradients Hunt et al. (2016). However, we observed in our setting that obtaining updates using the method followed in SeqGAN worked better.

## 3 ARCHITECTURE

Our proposed neural network does recurrent generation of speech samples. This generation is done by first encoding a speech sample to a rich latent variable. The latent variables are of two kinds. The first set of variables combine to form an embedding that captures the style and content of the speech sample, while the other set corresponds to discrete class variables that mark the accent of the

speech. The model does not need original samples during inference, and is capable of generating arbitrarily long samples starting from a random value for any given class of speech (i.e. the accent).

Figure 1 illustrates the architecture of our proposed approach, henceforth referred to as AccentGAN. It has an encoder layer which outputs latent variables $z$ and a decoder which transforms $z$ into output $y$. We would like $y$ to be distributed like the input $x$ and $z$ to be distributed according to a rich distribution. The former is enforced by a discriminator network, $C_\mu$ and the latter by a critic network $C_\nu$. Given a partial input, $x_{1:j}$, we shall compute $z_{1:T}$ where $z_{j+1:T}$ are sampled using a recurrent LSTM unit trained via policy gradients. Below, we describe each of these components in more detail.

## 3.1  ENCODER-DECODER

The encoder is a standard convolutional network that transforms an input speech sample to a constrained latent variable representation. The decoder is a deconvolution network with optional skip connections between the deconvolution layers and convolution layers of the encoder.

The loss function for the encoder (parameterized by $\theta$) and the decoder (parameterized by $\phi$) is defined as:

$$J_{\theta,\phi}(o, x; \mu) := \lambda ||o - x||_2 + (1 - C_\mu(o))^2 \tag{1}$$

where $C_\mu$ is the discriminator network, $x$ is an input speech segment, $o$ is a predicted output and $\lambda$ is a tunable hyperparameter. ($x$ and $o$ are both marked in Figure 1.)

## 3.2  DISCRIMINATOR

The discriminator network $C_\mu$ is a standard convolutional network, similar to the one used for encoding, but significantly deeper. The extra layers are added to make the discriminator more robust in differentiating between real and fake speech samples from the generator. We adopt the least squares loss function for the discriminator (Mao et al.). The loss function for the discriminator (parameterized by $\mu$) when the input segment is $x$ and the output from the decoder is $y$ is given as $J_{\mathsf{disc}}(C_\mu(x), C_\mu(y))$ where $J_{\mathsf{disc}}$ is defined as:

$$J_{\mathsf{disc}}(a, b) := ||\mathbf{1} - a||_2^2 + ||b||_2^2 \tag{2}$$

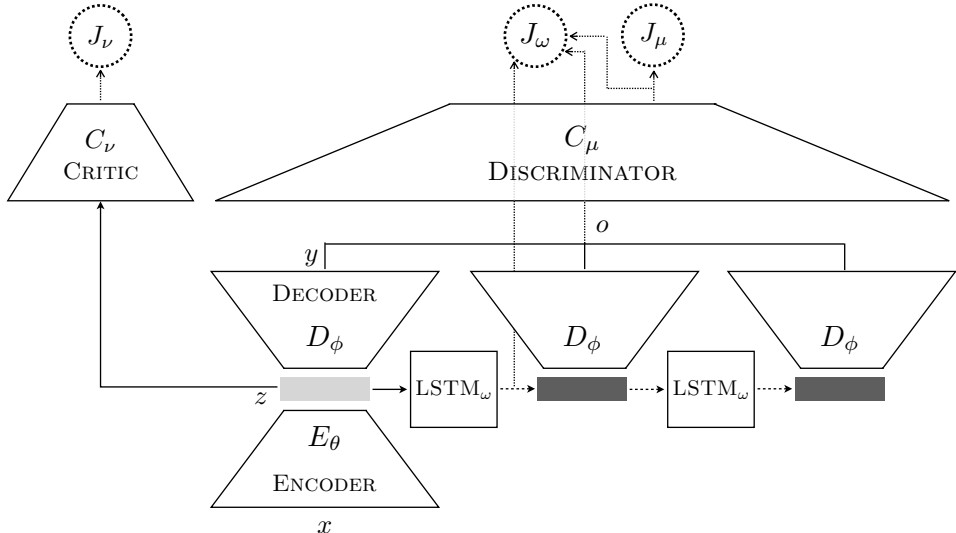

Figure 1: Schematic diagram illustrating the proposed GAN-based framework.

### 3.3 Distribution of the Latent Variables

The distribution of the latent variables $z$ used in our model are required to match a natural distribution. Specifically, the class variables are passed through a softmax layer. The rest of the latent variables are made to match a Gaussian distribution (Kingma & Welling, 2014). This is done by training a discriminant critic $C_\nu$ that is responsible for differentiating between latent variables that are being sampled from the associated distribution and the ones which are being generated from the encoder.

### 3.4 Recurrent Generation of Speech

During recurrent generation of a long speech sequence, we need to ensure correlation between the generated segments, which are individually generated in one time step of the architecture. Towards this, we incorporate a mechanism to force the sequence of latent variables $z_1, \ldots, z_T$ generated by encoding a speech sample $x_1, \ldots, x_T$ to be somewhat predictable from a prefix of the sequence. This mechanism involves a memory element (GRU or LSTM) that is trained to transform $z_j$ to $z_{j+1}$.

### 3.5 Reinforcement Learning Formulation

While the architecture so far may appear reasonable for generating speech, it turns out to perform very badly on its own. Speech sequences consist of a large number of frames $T$, and training the memory elements that output $z_1, \ldots, z_T$ directly using back-propagation seems quite ineffective, resulting in poor outputs.

To resolve this issue, we draw inspiration from SeqGAN (Yu et al., 2017), which modeled sequence generation in a GAN framework. The high-level idea is to use a reinforcement learning approach with delayed rewards to train the memory elements to produce a full valid sequence $z_1, \ldots, z_T$ from any prefix. In terms of the reinforcement learning framework, the *state* corresponds to the latent variables $z_j$; *actions* correspond to the transformation from $z_j$ to $z_{j+1}$ (alternately, the action can be viewed as producing the next segment of speech, as it is a deterministic function of $z_{j+1}$).[1] The *policy* is parametrized by the parameters $\omega$ of a memory element $\Psi_\omega$.

The gradients used to train the policy parameters $\omega$ are derived in Sutton et al. (1999), as:

$$\nabla_\omega J(\pi_\omega) = \mathbb{E}_{(s,a)}[Q^\pi(s,a)\nabla_\omega \log \pi_\omega(a|s)] \tag{3}$$

where $s$ is the state, $a$ is the action, and $Q^\pi(s,a)$ is the expected reward, and $\pi_\omega(a|s)$ is the probability assigned to action $a$ by the policy, given state $s$. As mentioned above, in our case, the state $s$ is the latent variable $z_j$ produced by the encoder after seeing a prefix $x_{1:j}$ of a segment $x_{1:T}$; the action is the next state $\hat{z}_{j+1}^{(j)}$ predicted using the LSTM network (with some added noise). The expected reward $Q^\pi(s,a)$ is replaced by the output of the discriminator $C_\mu(o^{(j)})$, where $o^{(j)}$ is a predicted output given $\hat{z}_{j+1}^{(j)}$: Note that the reward is higher if the discriminator is fooled more into considering $o^{(j)}$ to be from the real distribution. Here, instead of the expected reward, a single empirical sample of the prediction $o^{(j)}$ is used. The expectation over $(s,a)$ is also approximated by empirically summing over samples.[2] To predict $o^{(j)}$ we predict $\hat{z}_k^{(j)}$ for all $k > j$, using $\Psi_{\omega^*}$, where $\omega^*$ is the value of $\omega$ at the beginning of processing a batch. (This corresponds to an *off-policy* update of the parameter $\omega$.) Thus, we instantiate (3) as

$$\nabla_\omega J_\omega(z_j, \hat{z}_{j+1}^{(j)}, o^{(j)}; \mu, \omega) = \sum C_\mu(o^{(j)})\nabla_\omega \log \pi_\omega(\hat{z}_{j+1}^{(j)}|z_j) \tag{4}$$

where the summation is over all examples in a batch and all $j \in \{1, \ldots, T-1\}$, with the prediction $o^{(j)}$ computed as in Line 12 of Algorithm 1. As mentioned above we define $\hat{z}_{j+1}^{(j)} = \Psi_\omega(z_j) + \epsilon$,

---

[1]We remark that unlike SeqGAN, our actions were continuous. One may wonder if the SeqGAN framework is necessary, or appropriate, when the actions are continuous, and therefore admit training using back-propagation. But as discussed above, our experiments in fact show that the policy gradient approach of SeqGAN is useful even in this case.

[2]This could be viewed as a Monte Carlo approach of estimating $Q^\pi(s,a)$. An alternate approach using an actor critic model based on deterministic policy gradients Hunt et al. (2016) turned out to not perform as well in our setting.

---

**Algorithm 1** ACCENTGAN training pseudocode.

1: **Inputs**: data $D$, batch size $m$, segment length $T$, number of iterations $N_{\text{iter}}$, weight $\lambda$
2: Initialize $\phi, \mu, \theta, \omega, \nu$
3: **for** $i = 1$ to $N_{\text{iter}}$ **do**
4:     $\omega^* \leftarrow \omega$                                                      ▷ Make a working copy of the policy parameter
5:     Sample a batch of $m$ segments $x_{1:T}$ from data $D$
6:     **for** $j = 1$ to $T$ **do**
7:         $z_j \leftarrow E_\theta(x_j)$
8:         $y_j = D_\phi(z_j)$
9:         **for** $k = j + 1$ to $T$ **do**
10:             $\hat{z}_k^{(j)} = \pi_{\omega^*}(z_{k-1}) + \epsilon$ where $\epsilon \sim \mathcal{N}(0, \sigma^2 I)$
11:             $\hat{y}_k^{(j)} = D_\phi(\hat{z}_k^{(j)})$
12:         $o^{(j)} = [y_{1:j} | \hat{y}_{j+1:T}^{(j)}]$
13:         $\omega \leftarrow \text{update}(\omega, \nabla_\omega J_\omega(z_j, \hat{z}_{j+1}^{(j)}, o^{(j)}; \mu, \omega^*))$          ▷ Policy gradients in Eqn 5
14:         $\zeta \leftarrow \mathcal{N}(0, I)$
15:         $\nu \leftarrow \text{update}(\nu, \nabla_\nu J_{\text{disc}}(C_\nu(\zeta), C_\nu(z_j))$          ▷ $J_{\text{disc}}$ defined in Eqn 2
16:     $(\theta, \phi) \leftarrow \text{update}((\theta, \phi), J_{\theta,\phi}(o_T, x; \mu))$          ▷ $J_{\theta,\phi}$ defined in Eqn 1
17:     $\mu \leftarrow \text{update}(\mu, \nabla_\mu J_{\text{disc}}(C_\mu(x_{1:T}), C_\mu(y_{1:T}))$

---

where $\epsilon$ is an error vector (we consider all the vectors to be row vectors below). More precisely, we take $\epsilon \sim \mathcal{N}(0, \sigma^2 I)$. Then, we have $\pi_\omega(\hat{z}_{j+1}^{(j)} | z_j) \propto \exp(\frac{-1}{2\sigma^2} \|z_{j+1} - \Psi_\omega(z_j)\|_2^2)$. Then,

$$\nabla_\omega \log \pi_\omega(\hat{z}_{j+1}^{(j)} | z_j) = \nabla_\omega \frac{-\|\hat{z}_{j+1}^{(j)} - \Psi_\omega(z_j)\|_2^2}{2\sigma^2} = \frac{\hat{z}_{j+1}^{(j)} - \Psi_\omega(z_j)}{\sigma^2} \nabla_\omega \Psi_\omega(z_j^T).$$

Combined, our final expression for the gradient is

$$\nabla_\omega J_\omega(z_j, \hat{z}_{j+1}^{(j)}, o^{(j)}; \mu, \omega) = \sum C_\mu(o^{(j)}) \frac{(\hat{z}_{j+1}^{(j)} - \Psi_\omega(z_j))}{\sigma^2} \nabla_\omega \Psi_\omega(z_j^T). \tag{5}$$

Our complete training algorithm is outlined in Algorithm 1.

## 4 EXPERIMENTAL SETUP

We evaluate the efficacy of our proposed approach with the help of two standard datasets 1) The CSLU Foreign Accented English dataset (Lander, 2007) that contains accented speech in English from native speakers of 22 different languages and 2) The VCTK dataset (Veaux et al., 2010) that consists of speech from 109 speakers with varying accents. We used less than 5 hours of training data from both datasets. We trained our models using speech from 5-15 accents in the CSLU corpus and 20-30 speakers for the VCTK task.

AccentGAN was implemented using Tensorflow (Abadi et al., 2016). The encoder is a 12-layer convolution layer network with a kernel size of 31, while the decoder is a mirror image of the encoder network with the addition of skip connections. We use `Relu` (Arora et al.) units between the layers of the encoder network, while we use `LeakyRelu` (Arora et al.) units between the layers of the decoder network. We are able to fit up to 12 recurrent layers of the model in a single GPU while training. Skip connections are not used while generating samples.

The discriminator network has 2 more layers than the generator network giving it more discriminative power. The layers in this network again have a kernel width of 31 and we downsample by 2 in every layer. We also add batch normalization layers in order to ensure faster and more effective training (Ioffe & Szegedy).

The discriminator critic is a single-layer MLP with a large hidden layer containing 750 units that produces a single logit as output. The MLP layers are interspersed with `LeakyRelu` units. This network is trained using the least squares GAN loss function.

| Model | WGAN CRITIC | | Model | WGAN CRITIC |
|---|---|---|---|---|
| SpeechGAN | $346.21 \pm 3.21$ | | SpeechGAN | $276.42 \pm 15.02$ |
| SEGAN | $312 \pm 4.32$ | | SEGAN | $266.88 \pm 5.06$ |
| SampleRNN | $441.76 \pm 6.05$ | | SampleRNN | $412.88 \pm 13.45$ |
| Conditioned GAN | $291.16 \pm 3.2$ | | Conditioned GAN | $305.88 \pm 15.05$ |
| Policy GAN | $302.41 \pm 3.4$ | | Policy GAN | $337.34 \pm 20.04$ |
| AccentGAN | $201.98 \pm 7.8$ | | AccentGAN | $164.55 \pm 30.05$ |

Table 1: Wasserstein distances using critics trained on (a) the VCTK dataset (b) the CSLU dataset.

While doing inference and sampling from the trained model, we can either generate from noise for random content and use a random accent or start with a seed which is 1 second long and upsample to generate speech samples (that are at most 12 seconds in length). We use the latter in all our experiments.

## 5 RESULTS AND ANALYSIS

### 5.1 WGAN CRITIC BASED BENCHMARKING

As an objective evaluation metric, we use the Wasserstein distance from an independent Wasserstein GAN (Arjovsky et al., 2017) critic, trained on held out data, that is used to evaluate generated samples from our models. This measure has been adopted in prior work and is indicative of whether the generator is overfitting or whether the model is subject to mode collapse (Rosca et al., 2017; Danihelka et al., 2017). This WGAN critic is trained to maximize the following loss function (in the stochastic setting, with batch size $N$):

$$\mathcal{L}_{\text{CRITIC}} = \frac{1}{N} \left( \sum_{i=1}^{N} C_\mu(x_i^{\text{heldout}}) - \sum_{i=1}^{N} C_\mu(x_i^{\text{g}}) \right)$$

where $x^{\text{heldout}}$ corresponds to a batch of samples from the heldout set and $x^{\text{g}}$ is a batch of generated samples. We train the model by clipping weights but do not add the gradient penalty since large values of Wasserstein distances help us gain a better understanding of the model.

We build a first critic for the VCTK dataset that is trained on 50% of the held out data (corresponding to 23 speakers) and a second critic that is trained on the five most common accents appearing in the CSLU dataset. Table 1 shows Wasserstein distances on both datasets using both these critics. We compare AccentGAN with other models trained on exactly the same data. SpeechGAN is a naive model that generates speech using the standard GAN loss formulation, by reconstructing speech samples from noise. PolicyGAN and ConditionedGAN are two ablations of AccentGAN: the first variant does not involve conditioning of the latent variables and the second conditions the latent variable but does not use recurrent generation. SEGAN (Pascual et al., 2017) and SampleRNN (Mehri et al., 2017) are baseline models from prior work. We observe that AccentGAN performs the best cross both datasets and produces the smallest Wasserstein distance compared to the other models.

We also explore how the Wassertein distances change as we increase the number of accents that the critic is trained on. Figure 2 shows that the Wasserstein distance steadily drops with increasing number of accents.

### 5.2 HUMAN EVALUATION

The main objective of our model is to generate high-quality accented speech samples. In order to assess whether the speech accents are produced clearly in the generated samples and the speech is of high quality, we set up a human evaluation task to address both these questions. Six samples each from five different accents were generated using both AccentGAN and SEGAN.[3] We also used six ground-truth samples from each accent in the user study. Twelve pairs of samples, six samples

---

[3]We note here that we did not use SampleRNN in our user evaluation study since this system did not generate high quality speech clips, on account of the small amount of data that we used for training.

| Number of accents | Wasserstein distance |
|:---:|:---:|
| 5 | $708.45 \pm 5.1$ |
| 7 | $716.42 \pm 4.3$ |
| 10 | $704.41 \pm 4.1$ |
| 13 | $682.23 \pm 3.4$ |
| 15 | $672.43 \pm 5.2$ |

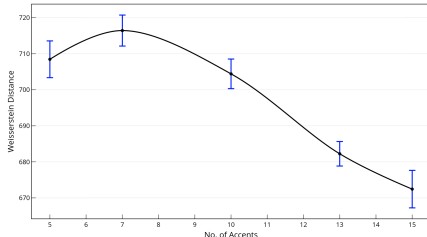

Table 2: (a)[CSLU dataset] Wasserstein distances using an alternate critic that is pre-trained on held-out data for 15 accents. This measures the average Wasserstein distance of samples generated by AccentGAN from ground truth samples, as we increase the number of accents we train on.
(b) [Graph] Represents the observed curve in the Wasserstein distances

each from two different accents, were presented to each user. In every pair, one was a ground-truth sample and the other came from either AccentGAN or SEGAN. No two utterances were identical in content across the twelve different pairs of samples. Users were asked first to listen to three reference samples in each accent to familiarize themselves with the underlying accent. Then, the users were asked to mark naturalness for both the samples in the pair on a 1-100 scale and they were also asked to mark on a numeric scale which of the two samples they thought was closer to the accent in the reference samples.

A total of 24 users participated in our study. The users were all undergraduate and graduate students and all of them were fluent speakers of English. Each combination of samples (AccentGAN vs. SEGAN, AccentGAN vs. ground-truth, SEGAN vs. ground-truth) got 2-3 scores from different users. Figures 2 and 3 show histograms of naturalness scores computed for AccentGAN vs. ground-truth samples and AccentGAN vs. SEGAN samples, respectively. (A score of 1.0 suggests that both systems were equally matched in naturalness scores, and scores greater and less than 1 suggest preferences for the respective systems.) We observe that for most pairs of speech samples, AccentGAN and ground-truth were equally matched. There were roughly equal number of speech samples that were rated higher for AccentGAN compared to ground-truth and vice-versa. For AccentGAN vs. SEGAN samples, however, we see a clear preference for AccentGAN samples in Figure 3.

Figure 4(a) shows a box plot of weighted accent preferences. The users assigned a continuous value between -50 and 50 for accent preference, which was bucketed into 5 classes and associated with a weight from $2, 1, 0, 1, 2$, thus giving more importance to samples which had a higher similarity score compared to samples with lower similarity scores. This re-weighting reduced all the values to the range -100 to 100. We see that preferences were fairly well-matched when AccentGAN samples were paired against ground-truth samples.

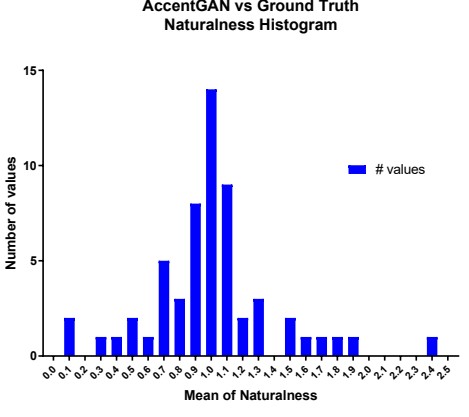

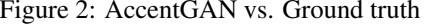

Figure 2: AccentGAN vs. Ground truth

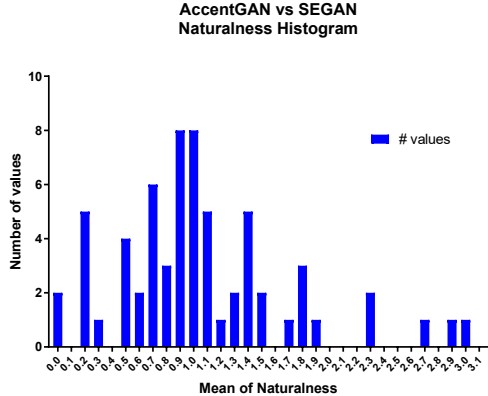

Figure 3: AccentGAN vs. SEGAN

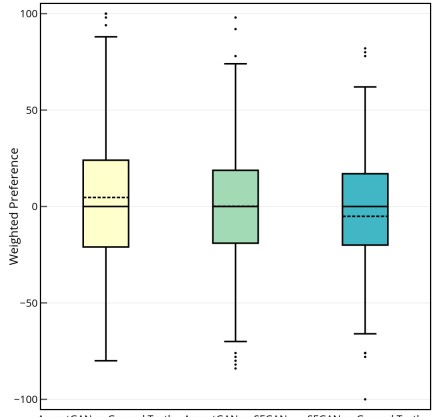

| Samples | Accuracy | Deviation |
|---|---|---|
| Ground Truth | 9.2% | 0.14 |
| AccentGAN | 9.3% | 0.08 |

| Model | L1 loss difference |
|---|---|
| Different Accents | 0.14 |
| AccentGAN | 0.023 |
| ConditionedGAN | 0.036 |

Figure 4: (**a** - Graph) Accent preference values from user evaluation study (**b** - Table upper) Accent recognition accuracies for ground truth vs. AccentGAN samples (**c** - Table lower) Difference between L1 losses for ground truth sample and accent transfer sample.

## 5.3 ACCENT RECOGNITION ACCURACY BASED EVALUATION

Accent recognition accuracy (ARA) is an alternate objective evaluation metric obtained using a trained accent identification system. We trained a classifier on speech samples generated for $N$ different accents and predict the accents of unseen generated samples. This is compared with a classifier trained on the same number of ground-truth speech samples from the same $N$ accents. We expect the ARA rates to be comparable across both systems, suggesting that our model generates accented speech samples that are very close in quality to the ground-truth samples.

To compute accent recognition accuracies, we extracted i-vector (Dehak et al., 2011) features from the speech samples; i-vectors are a low-dimensional representation of the speech signal that capture all relevant information about a speaker. i-vectors have been successfully used for accent identification in prior work (Bahari et al., 2013). We extracted 100-dimensional i-vectors using a background model that was trained on more than 100 hours of American accented speech (Godfrey & Edward, 1993) and we train a linear SVM classifier to predict the accent class of a test sample.

Figure 4(b) shows 3-fold accent recognition accuracies (and deviation statistics) on a test set of 130 samples (and training/validation sets of size 910 and 250, respectively). Each test sample was one of 15 accents. We observe very similar performance from using AccentGAN samples and ground-truth samples. We note here that the accent accuracies are low since we did not optimize the speech features and the classifier. Our objective was not to maximize ARA accuracy but to examine how AccentGAN samples fared against ground-truth samples in an accent recognition task.

## 5.4 TRANSFER LEARNING

We also experimented with transferring the content of speech across accents. This was done to ascertain whether the learned representations are (as claimed) disentangled in terms of accents and content. We also did this to generate samples with a specific accent and new content which appeared in other accents during training.

We passed batches of 64 samples each from the VCTK dataset into the auto-encoder and then changed the accent being conditioned for. We computed the true L1 loss between the sample generated by accent transfer via both AccentGAN and one ablation of this model, ConditionedGAN, as shown in Figure 4(c). We show that our model performs better giving a lower L1 loss value. As reference, we also show the true L1 loss between samples with the same content and different accents. (We used the VCTK dataset for this experiment as it has parallel corpora where the same content is rendered in different accents.)

## 5.5 COMPUTATIONAL EFFICIENCY

We compare our model against SampleRNN on two computational objectives: training time and inference time. We took roughly 6 hours to train AccentGAN on the CSLU dataset while SampleRNN took about 72 hours of training time. During inference, we could sample up to 10 samples of length 12 seconds each in one second of computation time on Tesla P-100 GPUs, which is an order of magnitude faster than SampleRNN.

We also notice that models such as SEGAN need to add more computationally intensive convolutional layers to generate longer samples, while we can generate arbitrary long samples by recurrently applying the autoencoder networks, given a rich enough seed. We successfully generated a speech sample which was 16 seconds long from 1 second of real speech.

## 6 CONCLUSION

We present a GAN-based approach for speech generation that efficiently generates long speech samples via policy gradients. We enable the generation of accented speech by explicitly conditioning on a latent variable that uses accent supervision. Future work includes extending this framework and conditioning on text to build text-to-speech synthesis systems.

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
