# OpenReview forum: "POLICY DRIVEN GENERATIVE ADVERSARIAL NETWORKS FOR ACCENTED SPEECH GENERATION"
_ICLR.cc/2018/Conference — Reject_

### Official Review · AnonReviewer1 · 2017-11-20
**Relevant work, but not executed or presented well**

**Rating:** 5
**Confidence:** 4

**Review:**

This paper presents a method for generating speech audio in a particular accent. The proposed approach relies on a generative adversarial network (GAN), combined with a policy approach for joining together generated speech segments. The latter is used to deal with the problem of generating very long sequences (which is generally difficult with GANs).

The problem of generating accented speech is very relevant since accent plays a large role in human communication and speech technology. Unfortunately, this paper is hard to follow. Some of the approach details are unclear and the research is not motivated well. The evaluation does not completely support the claims of the paper, e.g., there is no human judgment of whether the generated audio actually matches the desired accent.

Detailed comments, suggestions, and questions:
- It would be very useful to situate the research within work from the speech community. Why is accented modelling important? How is this done at the moment in speech synthesis systems? The paper gives some references, but without context. The paper from Ikeno and Hansen below might be useful.
- Accents are also a big problem in speech recognition (references below). Could your approach give accent-invariant representations for recognition?
- Figure 1: Add $x$, $y$, and the other variables you mention in Section 3 to the figure.
- What is $o$ in eq. (1)?
- Could you add a citation for eq. (2)? This would also help justifying that "it has a smoother curve and hence allows for more meaningful gradients".
- With respect to the critic $C_\nu$, I can see that it might be helpful to add structure to the hidden representation. In the evaluation, could you show the effect of having/not having this critic (sorry if I missed it)? The statement about "more efficient layers" is not clear.
- Section 3.4: If I understand correctly, this is a nice idea for ensuring that generated segments are combined sensibly. It would be helpful defining with "segments" refer to, and stepping through the audio generation process.
- Section 4.1: "using which we can" - typo.
- Section 5.1: "Figure 1 shows how the Wasserstein distance ..." I think you refer to the figure with Table 1?
- Figure 4: Add (a), (b) and (c) to the relevant parts in the figure.

References that might be useful:
- Ikeno, Ayako, and John HL Hansen. "The effect of listener accent background on accent perception and comprehension." EURASIP Journal on Audio, Speech, and Music Processing 2007, no. 3 (2007): 4.
- Van Compernolle, Dirk. "Recognizing speech of goats, wolves, sheep and… non-natives." Speech Communication 35, no. 1 (2001): 71-79.
- Benzeghiba, Mohamed, Renato De Mori, Olivier Deroo, Stephane Dupont, Teodora Erbes, Denis Jouvet, Luciano Fissore et al. "Automatic speech recognition and speech variability: A review." Speech communication 49, no. 10 (2007): 763-786.
- Wester, Mirjam, Cassia Valentini-Botinhao, and Gustav Eje Henter. "Are We Using Enough Listeners? No!—An Empirically-Supported Critique of Interspeech 2014 TTS Evaluations." In Sixteenth Annual Conference of the International Speech Communication Association. 2015.

The paper tries to address an important problem, and there are good ideas in the approach (I suspect Sections 3.3 and 3.4 are sensible). Unfortunately, the work is not presented or evaluated well, and I therefore give a week reject.

---

> ### Author Response · Authors · 2018-01-05
> **Human evaluation and Critic based evaluation is used keeping in mind the uniqueness of the problem being tackled**
>
> We thank the reviewer for all the valuable inputs.
>
> Motivation: The reviewer’s point about motivating the problem of generated accented speech further is well received and we thank the reviewer for pointing us to some relevant references. The revised version now contains a more detailed motivation.
>
> Accent-invariant representations for recognition: Our proposed approach could indeed be used to generate representations that can be incorporated within speech recognition systems for accented speech. This is a direction we intend to explore as future work and we consider this to be outside the scope of the current work.
>
> Human judgment of whether the generated audio matched the desired accent: We did actually conduct such a study. Section 4.2.2 describes the setup of our human evaluation study where we asked participants to listen to reference samples corresponding to a specific accent and then rate on a numeric scale how close a generated sample was to the accent in the reference sample.
>
> Effect of not adding structure to the latent representation: This is discussed in Table 2 which shows Wasserstein distances from an independent critic on different ablations of AccentGAN. PolicyGAN is identical to AccentGAN except there is no conditioning of the latent variables. We observe that PolicyGAN performs poorly in comparison to AccentGAN.
>
> The remaining comments on improving clarity will be addressed in the revised version.

---

> > ### Comment · AnonReviewer1 · 2018-01-08
> > **Human ratings of closeness of accent**
> >
> > Thanks for responding to the review.  I did spot in the original paper "they were also asked to mark on a numeric scale which of the two samples they thought was closer to the accent in the reference samples," but I could not find these result in the original paper nor in the revised version. Again, apologies if I am just missing these.

---

> > > ### Author Response · Authors · 2018-01-08
> > > **Human ratings of accent preference**
> > >
> > > Thanks for getting back to us. If you would look at Figure 4, the presented graph is the weighted preference (plotted to show the difference between models explicitly). We describe what exactly has been plotted in  the graph in section 5.2 (the last paragraph being most relevant). We decided not to report the actual values for space considerations, and hoped the representation would help us to get the point across better. We apologise for the confusion this might have caused.

---

> > > > ### Comment · AnonReviewer1 · 2018-01-09
> > > > **Reviewer response**
> > > >
> > > > Ah, thank you.  The figure caption is a bit confusing since you describe it as a "preference" rather than saying that you compare to a reference accent (as you do in the first par. of Section 5.2), but I think you have answered the question.

---

### Official Review · AnonReviewer2 · 2017-11-27
**The paper lacks any novel technical insight, contributions are not explained well, exposition is poor, and the evaluations are invalid.**

**Rating:** 3
**Confidence:** 4

**Review:**

The contributions made by this paper is unclear. As one of the listed contributions, the authors propose using policy gradient. However, in this setting, the reward is a known differentiable function, and the action is continuous, and thus one could simply backpropagate through to get the gradients on the encoder. Also, it seems the reward is not a function of the future actions, which further questions the need for a reinforcement learning formulation.

The paper is written poorly. For instance, I don't understand what this sentence means: "We condition the latent variables to come from rich distributions". Observed accent labels are referred to as latent (hidden) variables.

While the independent Wasserstein critic is useful to study whether models are overfitting (by comparing train/heldout numbers), their use for comparing across different model types is not justified. Moreover, since GAN-based methods optimize the Wasserstein distance directly, it cannot serve as a metric to compare GAN-based models with other models.

All of the models compared against do not use accent information during training (table 2), so this is not a fair comparison.

Overall, the paper lacks any novel technical insight, contributions are not explained well, exposition is poor, and the evaluations are invalid.

---

> ### Author Response · Authors · 2018-01-05
> **Independent Wasserstein Critics measure the distance between distributions well**
>
> The reviewer has made several objections. We agree to the extent that the exposition could have been improved. We would like to answer the other concerns below.
>
> Need for policy gradients: As we detailed in an answer to the first reviewer, simple back-propagation as the reviewer suggests demonstrably fails. Using policy gradients overcame the drawbacks of directly using back-propagation, without introducing significant computational overheads. While we had carried out extensive experimentation on this aspect, we omitted it entirely in our submission. We shall incorporate this in the revised version.
>
> Use of an independent Wasserstein critic to compare across models: We do not agree with the reviewer’s contention that using an independent Wasserstein critic to compare across models is unjustified. Not only is it a natural approach, but also one of the main uses detailed in  Danihelka et al. To quote from the paper: “If we use the independent critic, we can compare generators trained by other GAN methods or by different approaches.”
>
> Table 2 Comparisons:  The GAN models from the literature we compare with did not provide a means to incorporate accent information during training. Nevertheless, they were trained on data from a mix of accents identical to that in the validation/test data. So the additional data that our models were given corresponds to less than 5 bits per utterance, and this was essential for a harder task (of being able to generate speech in given accents) that is not captured in Table 2.
>
> We have tried to improve the presentation by removing some mysterious sounding phrasings (that resulted from using the vocabulary from our internal discussions). We apologize for any confusion they may have caused.
>
> We urge the reviewer to kindly reconsider their impression of the paper in light of our response.
>
> The remaining comments on improving clarity have been addressed in the revised version.

---

### Official Review · AnonReviewer3 · 2017-12-01
**Policy gradients are not needed for continuous latent variables**

**Rating:** 4
**Confidence:** 4

**Review:**

The paper considers speech generation conditioned on an accent class.
Least Squares GAN and a reconstruction loss is used to train the network.

The network is using continuous latent variables. These variables are trained by policy gradients.
I do not see a reason for the policy gradients. It would be possible to use the cleaner gradient from the discriminator.
The decoder is already trained with gradient from the discriminator.
If you are worried about truncated backpropagation through time,
you can bias it by "Unbiasing Truncated Backpropagation Through Time" by Corentin Tallec and Yann Ollivier.


Comments on clarity:
- It would be helpful to add x, z, y, o labels to the Figure 1.
I understood the meaning of `o` only from Algorithm 1.
- It was not clear from the text what is called the "embedding variable". Is it `z`?
- It is not clear how the skip connections connect the encoder and the decoder.
Are the skip connections not used when generating?
- In Algorithm 1, \hat{y}_k is based on z_k, instead of \hat{z}_k. That seems to be a typo.

Comments on evaluation:
- It is hard to evaluate speech conditioned just on the accent class.
Overfitting may be unnoticed.
You should do an evaluation on a validation set.
For example, you can condition on a text and generate samples
for text sentences from a validation set.
People can then judge the quality of the speech synthesis.
A good speech synthesis would be very useful.

---

> ### Author Response · Authors · 2018-01-05
> **Policy Gradients are efficient and robust during training, in comparison to standard backprop**
>
> We thank the reviewer for the valuable suggestions.
>
> Usefulness of policy gradients for continuous variables:
>
> We thank the reviewer for raising this question, as we should have included a discussion regarding this in the paper. (Indeed, given this question from two reviewers, a contribution of our work could be seen as showcasing the relevance of policy gradients even when the variables involved are continuous.)
>
> In the early stages of our project, we did experiment with plain back-propagation, as the reviewer suggested. But we observed that the resulting generated samples were of very poor quality. (We have uploaded a few samples from such a model at http://ec2-13-126-31-173.ap-south-1.compute.amazonaws.com:5000/ alongside samples generated by our proposed approach.) Hence we clearly needed techniques beyond plain back-propagation. Policy gradients appealed to us as we could readily adopt it to our setting, and immediately it gave us improvements over the original approach (with high quality utterances up to 12s long). Further, it did not add any significant computational overhead.
>
> We have added this discussion in the revised version.
>
> We do not deny the possibility that other recent approaches developed for similar purposes could also be adopted to our task, but the goal of this work has been to report the very significant improvements we achieved by adopting policy gradients.
>
>
> Comments on clarity: These have all been addressed in the revised version.
>
> Comments on evaluation: Conditioning on text and synthesizing accented speech is indeed part of our future work. Given the additional technical challenges involved, we have considered this to be outside the scope of the current work.
>
> The remaining comments on improving clarity have been addressed in the revised version.

---

### Decision · Program_Chairs · 2018-01-29
**ICLR 2018 Conference Acceptance Decision**

**Decision:**

Reject

**Comment:**

The paper proposes a method for accented speech generation using GANs.
The reviewers have pointed out the problems in the justification of the method (e.g. the need for using policy gradients with a differentiable objective) as well as its evaluation.